# Transverse Deflection for Extreme Ultraviolet Pellicles

**DOI:** 10.3390/ma16093471

**Published:** 2023-04-29

**Authors:** Sang-Kon Kim

**Affiliations:** The Faculty of Liberal Arts, Hongik University, Seoul 04066, Republic of Korea; sangkona@hongik.ac.kr

**Keywords:** computational lithography, extreme ultraviolet, EUV lithography, EUV mask, EUV pellicle, pellicle simulation, transverse deflection

## Abstract

Defect control of extreme ultraviolet (EUV) masks using pellicles is challenging for mass production in EUV lithography because EUV pellicles require more critical fabrication than argon fluoride (ArF) pellicles. One of the fabrication requirements is less than 500 μm transverse deflections with more than 88% transmittance of full-size pellicles (112 mm × 145 mm) at pressure 2 Pa. For the nanometer thickness (thickness/width length (t/L) = 0.0000054) of EUV pellicles, this study reports the limitation of the student’s version and shear locking in a commercial tool-based finite element method (FEM) such as ANSYS and SIEMENS. A Python program-based analytical-numerical method with deep learning is described as an alternative. Deep learning extended the ANSYS limitation and overcame shear locking. For EUV pellicle materials, the ascending order of transverse deflection was $Ru<{MoSi}_{2}=SiC<SiNx<{ZrSr}_{2}<p$-Si<Sn in both ANSYS and a Python program, regardless of thickness and pressure. According to a neural network, such as the Taguchi method, the sensitivity order of EUV pellicle parameters was Poisson’s ratio<Elastic modulus<Pressure<Thickness<Length.

## 1. Introduction

Up to the year 2000, extreme ultraviolet (EUV) pellicles were not considered for EUV masks. However, one of the problems for mass production of EUV lithography (EUVL) is defect control of EUV masks [1,2,3]. Using pellicles is a solution for defect control because pellicles as thin membranes can be placed above the masks to protect EUV masks from debris during the exposure process [4,5,6]. In addition, applying pellicles is economical because frequent particle inspection and cleaning of EUV masks are very expensive [7]. However, realistic application of full-size pellicles is disrupted due to fabrication requirements, which include a single transmittance of higher than 88~90%, uniformity of less than 0.4%, reflectance of less than 0.04%, mechanical deflection of less than 500 μm under a pressure load of 2 Pa, power enduring of higher than 400 W, $H^{+}$ durability, and a lifetime of longer than 315 h [8]. EUV transmission is directly related to pellicle thickness and EUVL performance because the EUV beam passes through a pellicle twice due to mask reflection. For a large silicon free-standing membrane, a thickness of less than 50 nm can obtain a high transmission of more than 80%. Thus, a 50 nm silicon membrane with wire-grids was researched by the Intel company [9]. Silicon nitride and polycrystalline pellicles have been developed due to their mechanical stability for a silicon-based membrane by the ASML company and the S&S Tech Corporation. Carbon nanotube (CNT) pellicles [10,11] with a porous structure and a nanometer-thick graphite film [12,13] as a monolayer-strong mechanical material have been developed by the IMEC company [14]. In particular, transverse deflection for EUV pellicles can locally influence pattern formations, such as critical dimension uniformity (CDU) and the lifetime of EUV masks [15]. Hence, understanding and analyzing transverse deflection of EUV pellicles are important for future EUV projection systems [16,17,18,19,20]. The cost of an EUV experiment is very high. In addition, EUVL processes are too complex to ignore simulation. However, accuracy and easy use of EUV pellicle simulation need improvement. As a research tool, EUV pellicle simulation can be used to perform experiments that otherwise would be very difficult or impossible with other methods. A pellicle simulation tool can be used to quickly evaluate options, optimize material properties, and/or save time and money by reducing the number of experiments in fab. Hence, pellicle simulations can be used to analyze, understand, and predict EUVL technologies for a gigabit era. Simulations of pellicles and contaminant particles in pellicles were performed to examine mechanical and thermal deformations and crack time following thermal deformation with a finite element method (FEM) tool such as ANSYS. However, there was no mention about design or solver limitation [21,22]. This study reported solver limitation and locking phenomenon in commercial software types, such as ANSYS and SIEMENS, for transverse deflection of EUV pellicles with a 10 mm×10 mm×54-nm (thickness/width length (t/L) = 0.0000054) structure. After the expected results of FEM and the theoretical results of an analytical-numerical method were compared with the expected experimental results, a Python program-based analytical-numerical method with deep learning as an alternative was described.

## 2. Simulation Methods

### 2.1. Mechanical Deformation

Mechanical deformation of pellicles mixes two effects, which are pellicle deformation by pressure and beam displacement by pellicle deformation. This deformation leads to defect patterns, including non-uniform CD due to non-uniform intensity distributions and image displacement.

Figure 1 presents a pellicle frame concept, a schematic sketch of a mask, and pellicle, and pellicle deflections without pressure and with pressure using ANSYS for a rectangular pellicle with clamped edges [23,24]. According to the Kirchhoff plate theory, a partial differential equation of transverse deflection (w) for a thin isotropic plate with pressure loading is(1)$$\nabla^{2}\left(D\nabla^{2}w\right)=q,D=\frac{Eh^{3}}{12\left(1-v^{2}\right)},$$
where D is bending stiffness of a plate, E is elasticity modulus, v is Poisson’s ratio, h is plate thickness, and q is pressure load. Many methods can be used to solve transverse deflection of thin plates. They can be divided into analytical methods, such as Timoshenko method [25], Galerkin-Vlasov method [26], Navier’s method [27,28], Levy’s method [29], numerical methods, such as finite element method (FEM) [30,31,32], finite boundary method (FBM) [33], global element method (GEM) [34], global-local finite element method (GLFEM) [35], and analytical-numerical methods [36]. Although analytical methods allow for the solving boundary problems of plates with unknown parameters and constrained by canonical contours, they are more accurate than numerical methods. However, numerical methods, such as FEM, allow us to solve the plate of arbitrary configuration approximately. Analytical-numerical methods overcome the disadvantages of other methods. One part of this method is performed in an analytical manner and the other with the help of numerical procedures.

### 2.2. Analytical-Numerical Method

Transverse deflection (w) of a plate in Equation (1) is solved using the sum of a general solution with pressure load q=0 and a particular solution:(2)$$w\left(x_{1},x_{2}\right)=R_{kpsv}W_{kpsv}\left(x_{1},x_{2}\right)+C_{mnpq}\cdot W_{mnpq}\left(x_{1},x_{2}\right),$$
where $x_{1}$ and $x_{2}$ are rectangular coordinates, $R_{kpsv}$ and $C_{mnpq}$ are coefficients from boundary conditions at plate contour, a shape function $W_{kpsv}\left(x_{1},x_{2}\right)$ of plate deflection is $B_{kpsv}\left(x_{s}\right)\cdot T_{kp\left(3-s\right)}\left(x_{3-s}\right)$, $B_{kpsv}$ is a basic function, $T_{kp\left(3-s\right)}\left(x_{3-s}\right)$ is a trigonometric function, $W_{mnpq}\left(x_{1},x_{2}\right)$ is a force function of plate deflection, and k, m, n, p, q, s, and v are indices. For a rectangular plate clamped at contour, boundary conditions are
(3)wx1,x2|x1=±a1=0; φ1x1,x2|x1=±a1=0;wx1,x2|x2=±a2=0; φ2x1,x2|x2=±a2=0,
where $\varphi_{1}$ and $\varphi_{2}$ are slopes of deflection [36]. For a particular solution of Equation (1), the distribution loaded on the upper surface of a plate is
(4)$$q\left(x_{1},x_{2}\right)=\frac{q_{0}}{D\left(\delta_{1}^{2}+\delta_{2}^{2}\right)}cos\left(\delta_{1}x_{1}\right)cos\left(\delta_{2}x_{2}\right),$$
where parameter $\delta_{s}$ is $\frac{\pi }{2a_{s}}$ for s = 1 or 2, $a_{s}$ is constant, and $q_{0}$ is load intensity.

### 2.3. Finite Element Method (FEM)

For finite element method (FEM), an approximate solution $\overset{-}{w}$ of w in Equation (1) can be expressed by piecewise linear functions which form a straight line in each subregion:(5)$$\overset{-}{w}\left(x_{j}\right)=\sum_{i=1}^{n+1}w_{i}N_{i}\left(x_{j}\right),\;\left(j=1,2\right)$$
where $w_{i}$ and $N_{i}\left(x_{j}\right)$ represent a value (w) and an interpolation (or shape) function in nodal points $i\left(=1,2,\ldots n,n+1\right)$, respectively. Galerkin method-based FEM adopts weighting functions $g_{i}\left(x_{j}\right)$ equal to interpolation functions $N_{i}\left(x_{j}\right)$. Coefficients of a solution (w) are determined to satisfy the following equation that error integral R($=\nabla^{2}\left(D\nabla^{2}w\right)-q$) in Equation (1) over an interest region weighted by arbitrary functions $g_{i}$ should be zero:(6)$$\int_{D}g_{i}Rdv=0,$$
where D is considered region [37].

### 2.4. Deep Learning

For a single variable method for linear regression, linear model functions $f_{w,b}\left(x^{\left(i\right)}\right)$ and cost functions $J\left(w,b\right)$ are, respectively,
(7)$$f_{w,b}\left(x^{\left(i\right)}\right)=wx^{\left(i\right)}+b,J\left(w,b\right)=\frac{1}{2m}\sum_{i=0}^{m-1}{\left(f_{w,b}\left(x^{\left(i\right)}\right)-y^{\left(i\right)}\right)}^{2},$$
where w and b are parameters. Gradient descent terms with a single variable are
(8)$$w=w-\alpha \frac{\partial J\left(w,b\right)}{\partial w},b=b-\alpha \frac{\partial J\left(w,b\right)}{\partial w},$$
(9)$$\frac{\partial J\left(w,b\right)}{\partial w}=\frac{1}{m}\sum_{i=0}^{m-1}\left(f_{w,b}\left(x^{\left(i\right)}\right)-y^{\left(i\right)}\right)x^{\left(i\right)},\frac{\partial J\left(w,b\right)}{\partial b}=\frac{1}{m}\sum_{i=0}^{m-1}\left(f_{w,b}\left(x^{\left(i\right)}\right)-y^{\left(i\right)}\right),$$
where m is number of training examples [38,39]. For a multiple variable method for linear regression, linear model functions $f_{w,b}\left(x^{\left(i\right)}\right)$ and cost functions $J\left(w,b\right)$ are, respectively,
(10)$$f_{w,b}\left(x^{\left(i\right)}\right)=w{\cdot x}^{\left(i\right)}+b,J\left(w,b\right)=\frac{1}{2m}\sum_{i=0}^{m-1}{\left(f_{w,b}\left(x^{\left(i\right)}\right)-y^{\left(i\right)}\right)}^{2},$$
where w and $x^{\left(i\right)}$ are vectors. Gradient descent terms with multiple variables are
(11)$$b=b-\alpha \frac{\partial J\left(w,b\right)}{\partial w},w_{j}=w_{j}-\alpha \frac{\partial J\left(w,b\right)}{\partial w_{j}},\;(j=1\ldots n-1)$$
(12)$$\frac{\partial J\left(w,b\right)}{\partial w_{j}}=\frac{1}{m}\sum_{i=0}^{m-1}\left(f_{w,b}\left(x^{\left(i\right)}\right)-y^{\left(i\right)}\right)x_{j}^{\left(i\right)},\frac{\partial J\left(w,b\right)}{\partial b}=\frac{1}{m}\sum_{i=0}^{m-1}\left(f_{w,b}\left(x^{\left(i\right)}\right)-y^{\left(i\right)}\right)$$
where n is number of features, $w_{j}\;and$ b are parameters, and $y^{\left(i\right)}$ is target value [40].

### 2.5. Response Surface Methodology

Response surface methodology (RSM) is a statistical approach to obtain an optimal response between two or more quantitative factors. As RSM, Taguchi method is a simple and efficient technique for identifying impacts of experimental parameters on system performance and computational cost by using signal-to-noise ratio (S/N) that is mean (signal) ratio to standard deviation (noise). S/N of lower-the-better (LB) criterion can be used to determine influential parameters for transverse deflection:(13)$$S/N=-10log\left(\frac{1}{n}\sum_{i=1}^{n}y^{2}\right),$$
where y and n are observed data and number of observations, respectively [41,42].

## 3. Results and Discussion

### 3.1. Plate Clamped at Contour

For an 8 m×4 m×0.2 m rectangular plate, results were obtained using a Python-based, analytical-numerical method [36] compared with the numerical solution obtained FEM in the ANSYS program (Student Edition 2021) and the SIEMENS program (Siemens NX-2000.3701), as shown in Table 1. The geometric and mechanical parameters in a Python program were a plate size of 8 m×4 m and a thickness of 0.2 m, Young’s modulus $E=30\times 10^{9}\;Pa$, Poisson’s ratio v=0.2, load pressure $q_{0}=10\;kPa$, and number of approximations K=5 (20 edge nodes). 

Figure 2 shows the maximum transverse deflections of an 8 m×4 m×0.2 m rectangular pellicle with clamped edges using a Python program, ANSYS, and SIEMENS. Figure 2a presents an 8 m×4 m×0.2 m rectangular plate. The ANSYS model consisted of 1272 nodes and 162 elements for a default mesh size, as shown in Figure 2c. SIEMENS model consisted of a 0.1 m mesh size, 36,165 nodes, and 6400 elements, as shown in Figure 2d. Results of calculations are presented in Table 1. According to Table 1, simulation results of a Python program were close to results of ANSYS and SIEMENS in a reasonable range. The ratio of thickness to width length $\left(t/L\right)$ was 0.025.

Figure 3 shows simulation results obtained using a Python program with an analytical-numerical method [36] for EUV pellicles with a 10 mm×10 mm×54-nm structure. Simulation conditions were Young’s modulus $E=169\times 10^{9}\;Pa$ and Poisson’s ratio v=0.22 for polycrystalline silicon (p-Si), Young’s modulus $E=454\times 10^{9}\;Pa$ and Poisson’s ratio v=0.25 for ruthenium (Ru), Young’s modulus $E=439.7\times 10^{9}\;Pa$ and Poisson’s ratio v=0.17 for molybdenum silicide ($Mo{Si}_{2}$), Young’s modulus $E=410\times 10^{9}\;Pa$ and Poisson’s ratio v=0.14 for silicon carbide (SiC), Young’s modulus $E=50\times 10^{9}\;Pa$ and Poisson’s ratio v=0.36 for tin (Sn), Young’s modulus $E=295\times 10^{9}\;Pa$ and Poisson’s ratio v=0.36 for zirconium disilicide ($Zr{Si}_{2}$), Young’s modulus $E=300\times 10^{9}\;Pa$ and Poisson’s ratio v=0.23 for silicon nitride (SiNx), number of approximations K=5 (20 edge nodes), and a pressure range of 50~200 Pa [21]. Figure 3a,c present the three-dimensional plots of the maximum transverse deflection for SiNx rectangular plates of 10 mm×10 mm×54-nm and 10 mm×10 mm×540-nm, respectively. Figure 3b,d show the plots of deflection vs. pressure for various materials at rectangular plates of 10 mm×10 mm×54-nm (thickness/with length (t/L)=0.0000054) and 10 mm×10 mm×540-nm (t/L=0.000054), respectively. The maximum transverse deflection was magnified 1000 times constantly due to thickness reduction of 0.1 multiplication. The difference in plot slopes due to various materials was similar regardless of the plate thickness.

In the case of ANSYS, as shown in Figure 4, the possible minimum thickness was designed at thickness t=5.4 μm. For a SiNx structure of 10 mm×10 mm×5.4 μm, Figure 4a,b present a schematic mesh and maximum transverse deflection at 50 Pa pressure, respectively. Simulation conditions were the same as those in Figure 3. Figure 4c,d shows the plots of maximum transverse deflection vs. pressure due to various materials at thicknesses t=5.4 μm (t/L=0.00054) and t=54 μm (t/L=0.0054), respectively. Although a slope of 5.4 μm thickness was slightly curved in Figure 4c, the difference in curve slopes due to various materials was similar regardless of the pellicle thickness. ANSYS (Student Edition 2021) has limitations:Minimum mesh size of more than 100 μm size;Meshing of more than 10 μm thickness. A structure of 10mm×10mm×540-nm did not mesh;Design of more than 10-nm scale;Solving of less than 20,000 mash nodes.

### 3.2. Results of Deep Learning

A single variable method for linear regression in deep learning was used to overcome ANSYS limitation using a Python program [43]. Table 2 shows the expected ANSYS results (${log}_{10}(deflection))$ for structures of 10 mm×10 mm×54-nm and 10 mm×10 mm ×540-nm. After 10,000 iterations at a pressure of 50 Pa, from Equations (8) and (9), cost function $J\left(w,b\right),$ $\partial J\left(w,b\right)/\partial b$, $\partial J\left(w,b\right)/\partial w$, w, and b were 0.0346, 0.02876, 0.06512, −2.342, and −17.0814, respectively.

Performance of gradient descent was improved by feature scaling using calculation of ${log}_{10}$. Expected transverse deflections of ANSYS were calculated at thicknesses of 54-nm and 540-nm using four methods of deep learning, which are a Python program, 1-layer TensorFlow [44], 2-layer TensorFlow [45], and a machine learning toolkit called Scikit-Learn [46]. Figure 5 shows a comparison of expected values of four methods to ANSYS results. For 1-layer TensorFlow, after 10,000 iterations at a pressure of 50 Pa, cost function $J\left(w,b\right)$, w, and b were 0.0434, −2.5313423, and −17.91587, respectively. For 2-layer TensorFlow, after 10,000 iterations at a pressure of 50 Pa, cost function $J\left(w,b\right)$ was 0.0434, parameters $w_{1}$, $w_{2}$, $b_{1}$, and $b_{2}$ at 1-layer were −1.1386404, 0.0460223, −5.820591, and 0.25007814, respectively. Parameters $w_{1}$, $w_{2}$, and b in 2 layer were 2.2235708, −0.0971202, and −4.970218, respectively. For Scikit-Learn, after 10,000 iterations at a pressure of 50 Pa, parameters w and b were −2.54 and −17.937115, respectively. Expected values of the four deep learning methods were in agreement with results obtained using ANSYS in a reasonable range, as shown in Figure 5.

Figure 6a presents a comparison of the simulation results of ANSYS in Figure 4 and a Python program in Figure 3. The difference in ${log}_{10}(deflection)$ values due to pressure was increased at thickness t=5.4 μm (t/L = 0.00054), as shown in Figure 6a. For transverse deflections obtained from deep learning of a Python program at t=54-nm (t/L = 0.0000054) and t=540-nm (t/L = 0.000054), the difference was further increased. In other words, the results of FEM using ANSYS were different from those of an analytical-numerical method using a Python program, when t/L was more minor than 0.0054. According to experiments of transverse deflections in Ref. [22], low-pressure chemical vapor deposition (LPCVD) was used to deposit a 60 nm thick silicon nitride layer onto a (100) 725 μm thick silicon wafer using dichlorosilane (SiH_2_Cl_2_) and ammonia (NH_3_). Free-standing silicon nitride membranes with dimensions of 10 mm × 10 mm were fabricated by silicon back-side etching at 60 °C in a potassium hydroxide (KOH) and isopropyl alcohol mixture solution. A displacement sensor and pressure sensor had measurement resolutions of 1 μm and 1 Pa, respectively [47]. There was a large difference (~$10^{6}$) between theoretical results and expected experimental results (red dots) based on experimental results in Ref. [22], as shown in Figure 6b. One of the reasons is shear loading, an error that occurs in FEM due to the effect on stiffness matrix as the length-to-thickness ratio in beam element.

According to a scenario, if a beam with thickness t is under combined shear deformation of a function t and bending deformation of a function $t^{3}$, $t^{3}$ approaches zero much faster than *t* when t is very small. A beam’s strain energy will come from shear deformation [48,49,50]. Thus, this can lead to erroneous results for very thin beams. The difference between results of FEM using ANSYS and an analytical-numerical method using a Python program in Figure 6a,b can be attributed to locking phenomena, such as shear locking. The smaller the thickness/length (t/L), the greater the error. In addition, shear locking can cause a difference (~$10^{6}$) between the theoretical results included in the FEM and an analytical-numerical method and the expected experimental results. A multiple variable method for linear regression in deep learning was used to overcome shear loading using a Python program and Scikit-Learn. Table 3 shows the expected results (${log}_{10}(deflection))$ of deep learning for a 10mm×10mm×54-nm structure. After 100,000 iterations using a Python program (and Scifit-Learn), parameters $w_{1}$, $w_{2}$, and b were −0.00033 (and −0.000425), 0.658 (and 0.677), and −5.619978 (and −5.654055) from Equations (8) and (9), respectively. Expected values of a Python program and Scikit-Learn were in agreement with the expected experimental data, as shown in Table 3.

### 3.3. Anaysis

Figure 7 presents maximum transverse deflections in rectangular pellicles with clamped edges due to various materials using ANSYS and a Python program. Regardless of thickness and pressure, the ascending order of transverse deflections due to pellicle materials was $Ru<{MoSi}_{2}=SiC<SiNx<{ZrSr}_{2}<p$-Si<Sn in ANSYS and a Python program. Figure 8 shows the sensitivity of EUV pellicle’s parameters for transverse deflection using a surface response analysis, such as Taguchi method in Minitab^TM^, a commercial tool. EUV pellicle parameters’ plots present signal-to-noise ratio (S/N) in Equation (13). Red arrows as plot slopes are the optimal setting of EUV pellicle’s parameters to minimize transverse deflection. According to a neural method, pellicle length has the most significant impact on EUV pellicle deflection, followed by thickness, pressure, elastic modulus, and Poisson’s ratio. In order to reduce transverse deflection, length and pressure should be smaller, and thickness, elastic modulus, and poison’s ratio should be larger.

## 4. Conclusions

Since pellicles have been a solution for defect control of EUV masks, theoretical approaches for EUV pellicles have become necessary for pellicle analysis. In this paper, transverse deflections of EUV pellicles were simulated using a Python program, ANSYS, and SIEMENS. The possible minimum thickness solved using ANSYS was 5.4 μm with a rectangular plain of 10 mm × 10 mm due to the student’s version limitation. For ANSYS limitation, expected values of single variable methods for linear regression were in agreement with the results obtained using ANSYS in a reasonable range. The maximum transverse deflection in an analytical-numerical method was magnified 1000 times constantly due to a thickness reduction of 0.1 multiplication. The difference of transverse deflection due to pressure between results of FEM and an analytical-numerical method was increased at thickness *t* = 5.4 μm (*t*/*L* = 0.00054). For the difference (~$10^{6}$) between expected experimental results and theoretical results that included FEM using ANSYS and an analytical-numerical method using a Python program due to shear locking, expected values of multiple variable methods for linear regression were in agreement with the expected experimental data. Hence, a Python program-based analytical-numerical method with deep learning is an alternative way to analyze EUV pellicles. Regardless of thickness and pressure, the ascending order of transverse deflections due to pellicle materials was $Ru<{MoSi}_{2}=SiC<SiNx<{ZrSr}_{2}<p$-Si<Sn in ANSYS and a Python program. According to a neural network, the ascending order of sensitivity of parameters on traverse deflection was Poison’sratio<Elasticmodulus<Pressure<Thickness<Length.

## Figures and Tables

**Figure 1 materials-16-03471-f001:**
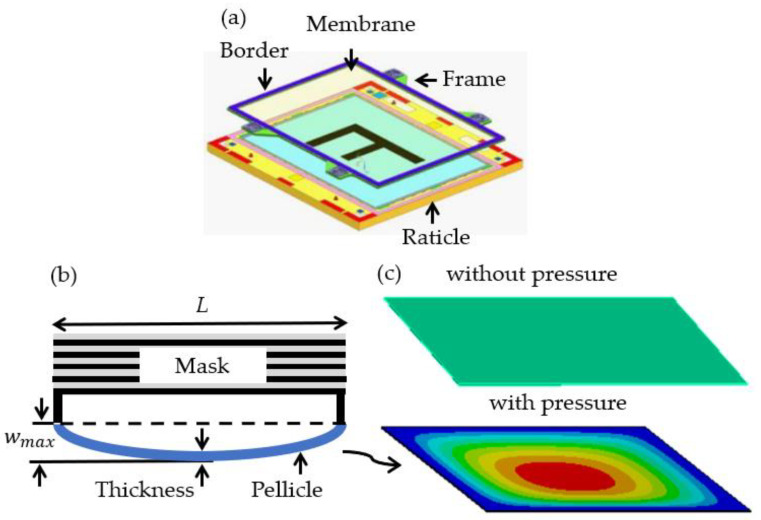
Schematic sketch of transverse deflection in a rectangular pellicle with clamped edges due to pressure: (**a**) a pellicle frame concept [20], (**b**) a schematic sketch of a mask and pellicle, and (**c**) schematic pellicles without pressure and with pressure using ANSYS. L is width length of a pellicle and $w_{max}$ is maximum transverse deflection of a pellicle.

**Figure 2 materials-16-03471-f002:**
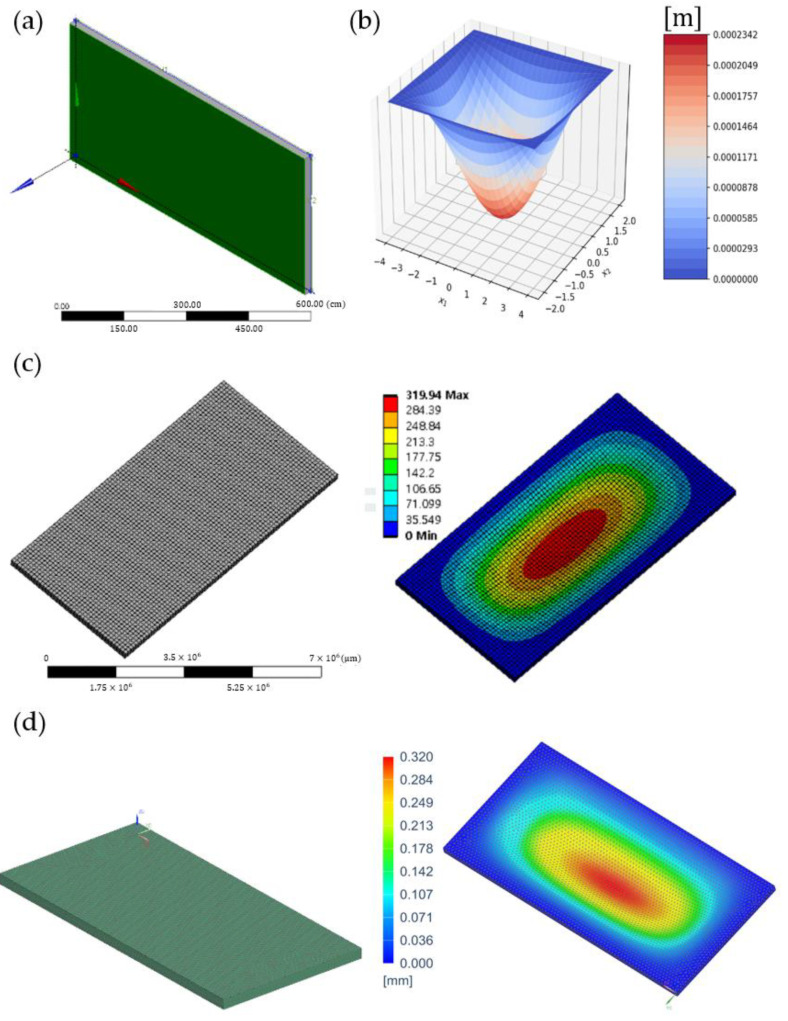
Maximum transverse deflections in a rectangular pellicle with clamped edges: (**a**) a 8 m×4 m×0.2 m plate structure, (**b**) a three-dimensional plot of deflection using a Python program, and plots of the schematic mesh and the transverse deflection contours using (**c**) ANSYS and (**d**) SIEMENS, respectively.

**Figure 3 materials-16-03471-f003:**
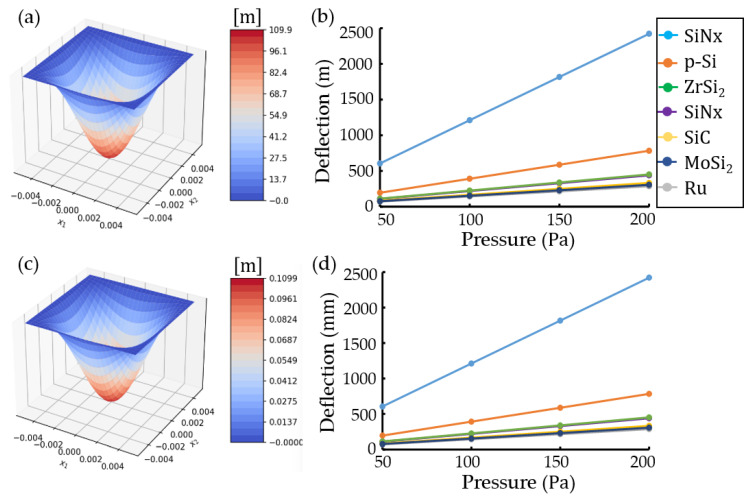
Maximum transverse deflections in a rectangular pellicle *of*
10 mm×10 mm×thickness(t) with clamped edges using a Python program: (**a**,**c**) three-dimensional contours of deflection for silicon nitride (SiNx) rectangular plates at thickness t=54-nm and 540-nm, and (**b**) and (**d**) plots of deflection vs. pressure at t=54-nm and 540-nm, respectively. The thickness ratio of width length (t/L) is 0.0000054 at t=54-nm and t/L is 0.000054 at 540-nm.

**Figure 4 materials-16-03471-f004:**
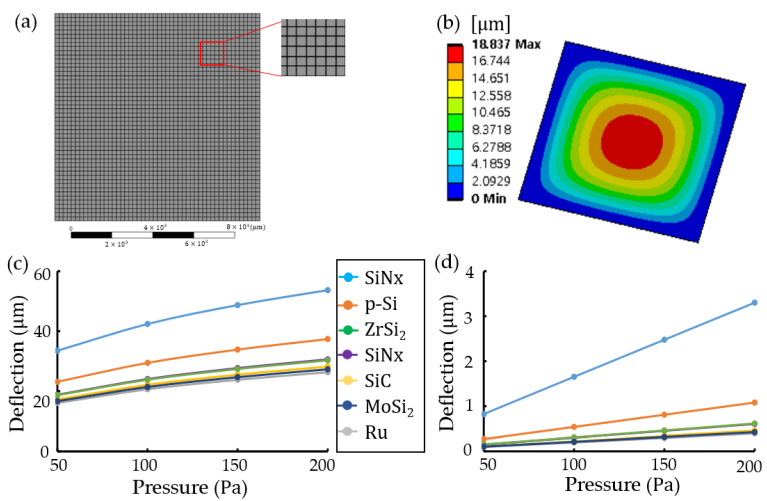
Maximum transverse deflections in a rectangular pellicle (10 mm×10 mm×thickness(t)) with clamped edges using ANSYS: for a silicon nitride (SiNx) structure of 10 mm×10 mm×5.4 μm, (**a**) a schematic mesh and (**b**) a transverse deflection contour at a pressure of 50 Pa, and for various materials, plots of maximum transverse deflection vs. pressure at (**c**) t=5.4 μm(t/L=0.00054) and (**d**) t=54 μm(t/L=0.0054).

**Figure 5 materials-16-03471-f005:**
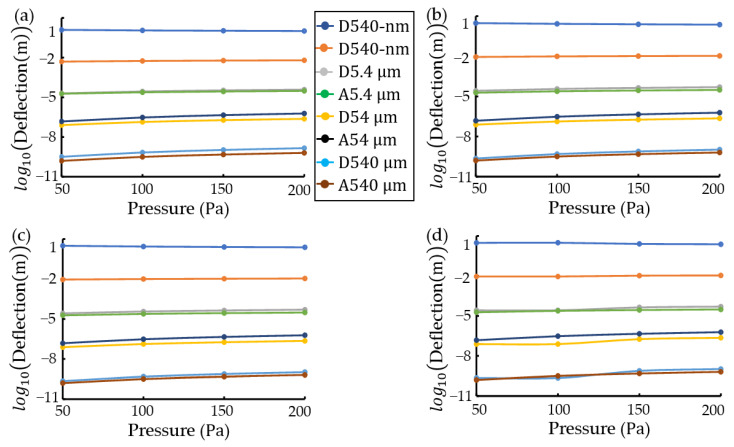
Plots of ${log}_{10}(deflection)$ vs. pressure due to thickness to compare expected values of deep learning to ANSYS results: expected transverse deflections using (**a**) a Python program, (**b**) 1-layer TensorFlow, (**c**) 2-layer TensorFlow, and (**d**) Scikit-Learn as a machine learning toolkit. ‘D’ and ‘A’ mean deep learning and ANSYS, respectively.

**Figure 6 materials-16-03471-f006:**
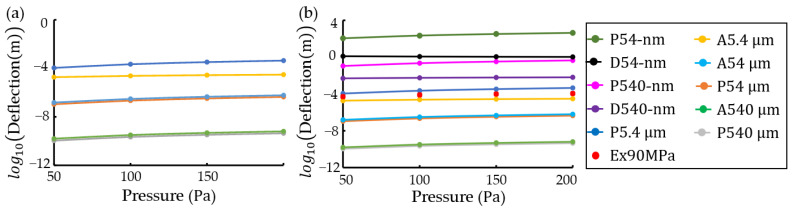
Plots of ${log}_{10}(deflection)$ vs. pressure to compare expected values of a Python program and ANSYS results for a SiNx pellicle of a 10 mm×10 mm×54-nm structure: (**a**) before deep learning and (**b**) after deep learning. ‘P’, ‘A’, ‘D’, and ‘Ex90MPa’ mean a Python program, ANSYS, deep learning, and expected experimental results with 90 MPa residual stress, respectively.

**Figure 7 materials-16-03471-f007:**
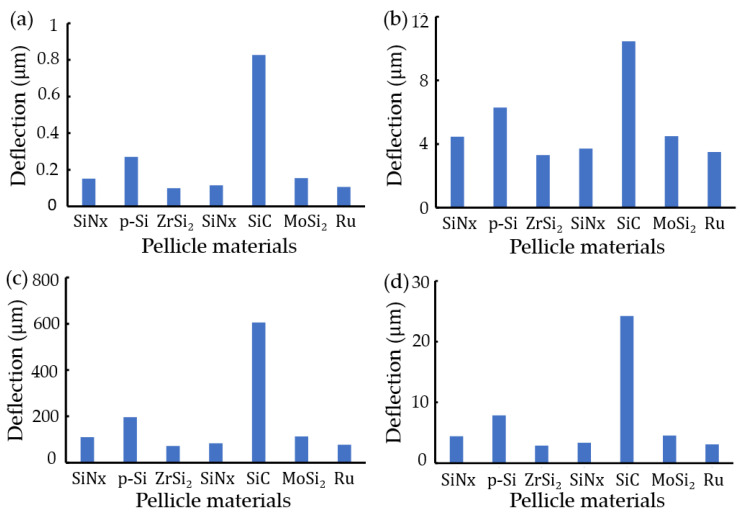
Maximum transverse deflections of rectangle pellicles with clamped edges due to various materials: ANSYS results in (**a**) a 10 mm×10 mm×54 μm structure at 50 Pa pressure and (**b**) a 10 mm×10 mm×5.4 μm structure at 2 Pa pressure, results of a Python program in (**c**) a 10 mm×10 mm×54-nm at 50 Pa pressure, and (**d**) a 10 mm×10 mm×5.4 μm structure at 2 Pa pressure.

**Figure 8 materials-16-03471-f008:**
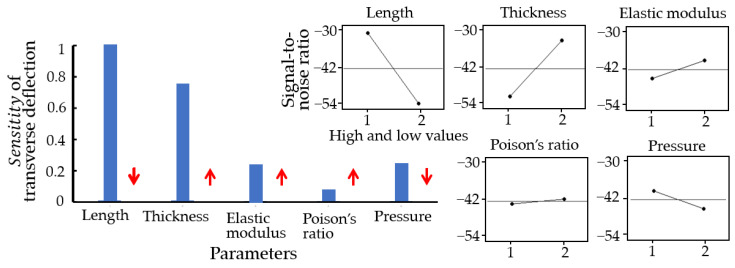
Sensitivity of EUV pellicle’s parameters, which are length, thickness, elastic modulus, Poison’s ratio, and pressure. Parameter sensitivities are normalized by the most sensitive parameter, length. EUV pellicle parameters’ plots and red arrows present signal-to-noise ratio (S/N) and plot slopes, respectively.

**Table 1 materials-16-03471-t001:** Minimal and maximal values of transverse deflection.

	Deflection	
Unit: m	Min	Max
Python program	0.0	0.0002343
ANSYS	0.0	0.00031428
SIEMENS	0.0	0.00032994

**Table 2 materials-16-03471-t002:** Expected ANSYS results (${log}_{10}(deflection))$ for structures of 10 mm×10 mm×54-nm and 10 mm×10 mm×540-nm using a Python program.

	50 Pa	100 Pa	150 Pa	200 Pa
54-nm	0.099612	0.058243	0.032225	0.013248
540-nm	−2.298013	−2.241721	−2.218677	−2.198874

**Table 3 materials-16-03471-t003:** Comparison of experimental results (${log}_{10}(deflection))$ and expected results of deep learning using a Python program and Scikit-Learn for a 10mm×10mm×54-nm structure.

	50 Pa	100 Pa	150 Pa	200 Pa
Exp. ^1^	−4.29243	−4.110698	−4.010995	−3.948847
Python	−4.293415	−4.111958	−4.012543	−3.94681
Scikit	−4.292584	−4.110012	−4.011903	−3.94847

^1^ Expected experimental data at 90 MPa residual stress.

## Data Availability

The data required to reproduce are available to obtain from the corresponding author.

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
