# Peer review of "Transverse Deflection for Extreme Ultraviolet Pellicles"

_materials, 2023, doi:10.3390/ma16093471_

Round 1
Reviewer 1 Report
Accept in present form.
Author Response
(Answer) Thank you. This paper was revised two times by the company of written English. This is certificate of editing.

Reviewer 2 Report
The author presents a study of transverse deflections of pellicules based on finite element method for solving the deflection of a rectangular plate based on Kirchhoff theory, and compare the results with analytical results as implemented in a Python program with deep-learning. While the theme is somewhat interesting, the main results obtained suggests that the FEM student version algorithms are limited when compared to an analytical solution, which should be expected. Still, the author provides a thorough analysis of this aspect of the problem based on numerous results, so this article can be accepted for publication after some details are accounted for.
1- The written english is very difficult to comprehend in many passages of the text, so I recommend a revision of the text by a native english speaker.
2- The introduction section should be enriched with more context about the problem. The way I see it, this introduction is acceptable as is, however the article quality would benefit from a more mindful writing.
3- In the conclusions it is stated that the obtained results could provide theoretical insights for the development of new materials and fabrication procedures to enhance performance of pellicules, however it is not clear how these results could be employed in this way. It would be a good idea to give this context in the introduction and clarify this kind of idea.
4- It is not clear why a FEM solution would be necessary for this problem, while the analytical solutions provide much more reliable data, even in agreement with the experimental results. The motivations to develop this kind of comparison should be stated more clearly.
5- The figures that present the obtained results are not elaborated carefully. The content of some figures is very difficult to read, and special care must be adopted in order to avoid the theoretical curves to cross the curve labels, as in figures 3b and 4c. Also, it seems that the units of transverse deflection are wrong in figure 3.
6- Around equation 1, the quantity w is not defined explicitly.
My recommendation is that this work is to be accepted for publication after all of these points are addressed satisfactorily.
Author Response
1- The written english is very difficult to comprehend in many passages of the text, so I recommend a revision of the text by a native english speaker.
(Answer) Thank you. This paper was revised two times by the company of written English. This is certificate of editing.

2- The introduction section should be enriched with more context about the problem. The way I see it, this introduction is acceptable as is, however the article quality would benefit from a more mindful writing.
(Answer) Thank you. Thank you. Sentences were added.
In 45th line in introduction,
The cost of an EUV experiment is very high. In addition, EUVL processes are too com-plex to ignore simulation. However, accuracy and easy use of EUV pellicle simulation need improvement. As a research tool, EUV pellicle simulation can be used to perform experiments that otherwise would be very difficult or impossible with other methods. A pellicle simulation tool can be used to quickly evaluate options, optimize material properties, and/or save time and money by reducing the number of experiments in fab. Hence, pellicle simulations can be used to analyze, understand, and predict EUVL technologies for a gigabit era. Simulations of pellicles and contaminant particles in pellicles were performed to examine mechanical and thermal deformations and crack time following thermal deformation with a finite element method (FEM) tool such as ANSYS. However, there was no mention about design or solver limitation.
In 58th line in introduction,
After expected results of FEM and theoretical results of an analytical-numerical method were compared with experimental results,
3- In the conclusions it is stated that the obtained results could provide theoretical insights for the development of new materials and fabrication procedures to enhance performance of pellicules, however it is not clear how these results could be employed in this way. It would be a good idea to give this context in the introduction and clarify this kind of idea.
(Answer) The last sentence in conclusion was deleted and two sentences were added.
In 292th line in conclusion,
The maximum transverse deflection in an analytical-numerical method was magnified 1000 times constantly due to thickness reduction of 0.1 multiplication. The difference of transverse deflection due to pressure between results of FEM and an analytical-numerical method was increased at thickness t= 5.4 μm (t/L = 0.00054).
4- It is not clear why a FEM solution would be necessary for this problem, while the analytical solutions provide much more reliable data, even in agreement with the experimental results. The motivations to develop this kind of comparison should be stated more clearly.
(Answer) Thank you.
One of motivations is that simulations of pellicles and contaminant particles in pellicles were performed for mechanical and thermal deformations and crack times by a finite element method (FEM) tool, such as ANSYS. However, there was no mention about design or solver limitation. This study reported solver limitation and locking phenomenon in commercial software types, such as ANSYS and SIEMENS, for transverse deflection of EUV pellicles with a 10 mm ×10 mm ×54-nm (thickness/width length (t⁄L) = 0.0000054) structure.
In 45th line in introduction,
The cost of an EUV experiment is very high. In addition, EUVL processes are too com-plex to ignore simulation. However, accuracy and easy use of EUV pellicle simulation need improvement. As a research tool, EUV pellicle simulation can be used to perform experiments that otherwise would be very difficult or impossible with other methods. A pellicle simulation tool can be used to quickly evaluate options, optimize material properties, and/or save time and money by reducing the number of experiments in fab. Hence, pellicle simulations can be used to analyze, understand, and predict EUVL technologies for a gigabit era. Simulations of pellicles and contaminant particles in pellicles were performed to examine mechanical and thermal deformations and crack time following thermal deformation with a finite element method (FEM) tool such as ANSYS. However, there was no mention about design or solver limitation.
5- The figures that present the obtained results are not elaborated carefully. The content of some figures is very difficult to read, and special care must be adopted in order to avoid the theoretical curves to cross the curve labels, as in figures 3b and 4c. Also, it seems that the units of transverse deflection are wrong in figure 3.
(Answer) Thank you. All of figures are redrawn. The [m] unit of figure 3 is not wrong. After showing figures 3 and 4, the following figures are simulation results by using a Python program.
Figure 3,

Figure 4,

The [m] unit of figure 3 is not wrong. The following figures are simulation results by using a Python program. For SiNx material and a 10 mm x 10 mm x 54 nm structure.

The maximum transverse deflection in an analytical-numerical method was magnified 1000 times constantly due to thickness reduction of 0.1 multiplication.
6- Around equation 1, the quantity w is not defined explicitly.
(Answer) Thank you.
In 75th line
a partial differential equation of transverse deflection (w)

Reviewer 3 Report
My research is on nanofabrication, and I am familiar with EUV lithography as I have been teaching it over the past many years. But I am NOT familiar with numerical simulation or programming. Thus I cannot judge whether the simulation results in this manuscript are publishable or not.
My main concern is the novelty of this research work. The introduction does not mention what had been done in this field, what is new for the current work, and what are the advantage of the current methods as compared to previous works. The author must describe clearly the novelty of the work before it can be published.
Author Response
My main concern is the novelty of this research work. The introduction does not mention what had been done in this field, what is new for the current work, and what are the advantage of the current methods as compared to previous works. The author must describe clearly the novelty of the work before it can be published.
(Answer) Thank you. The following sentences describe the novelty of this work:
- Simulations of pellicles and contaminant particles in pellicles were performed for mechanical and thermal deformations and crack times by a finite element method (FEM) tool, such as ANSYS. However, there was no mention about design or solver limitation. This study reported solver limitation and locking phenomenon in commercial software types, such as ANSYS and SIEMENS, for transverse deflection of EUV pellicles with a -(thickness/width length () = 0.0000054) structure.
- A deep learning method was used for ANSYS limitation of design and solver for nano-thickness. An analytical-numerical method as an alternative way was introduced.
- After expected results of FEM and theoretical results of an analytical-numerical method were compared with the experimental results, a deep learning modeling was used for the complement of the difference.
- Hence, this paper gives guidelines of using commercial tools, such as ANSYS and SIEMENS for better usage.
- This paper suggests a method, such as a Python program based analytical-numerical method with deep learning, to analyze structures with less than thickness/width length () = 0.0000054.
The following paragraphs are added in this paper.
In 45th line in introduction,
The cost of an EUV experiment is very high. In addition, EUVL processes are too com-plex to ignore simulation. However, accuracy and easy use of EUV pellicle simulation need improvement. As a research tool, EUV pellicle simulation can be used to perform experiments that otherwise would be very difficult or impossible with other methods. A pellicle simulation tool can be used to quickly evaluate options, optimize material properties, and/or save time and money by reducing the number of experiments in fab. Hence, pellicle simulations can be used to analyze, understand, and predict EUVL technologies for a gigabit era. Simulations of pellicles and contaminant particles in pellicles were performed to examine mechanical and thermal deformations and crack time following thermal deformation with a finite element method (FEM) tool such as ANSYS. However, there was no mention about design or solver limitation.
In 58th line in introduction,
After expected results of FEM and theoretical results of an analytical-numerical method were compared with experimental results,
In 292th line in conclusion,
The maximum transverse deflection in an analytical-numerical method was magnified 1000 times constantly due to thickness reduction of 0.1 multiplication. The difference of transverse deflection due to pressure between results of FEM and an analytical-numerical method was increased at thickness t = 5.4 μm (t/L = 0.00054).
Reviewer 4 Report
In this manuscript, transverse deflections of EUV pellicles were simulated using a Python program, ANSYS, 272 and SIEMENS. This study pro- vides theoretical insights into developing new pellicle materials and fabrication for achieving a better performance. The topic is interesting and the manuscript is written well. It can be considered for publication in the present form. It is suggested the authors modify Fig 2 to increase its resolution.
Author Response
In this manuscript, transverse deflections of EUV pellicles were simulated using a Python program, ANSYS, 272 and SIEMENS. This study pro-vides theoretical insights into developing new pellicle materials and fabrication for achieving a better performance. The topic is interesting and the manuscript is written well. It can be considered for publication in the present form. It is suggested the authors modify Fig 2 to increase its resolution.
(Answer) Thank you. Figure 2 was redrawn.

For the written English
(Answer) Thank you. This paper was revised two times by the company of written English. This is certificate of editing.

Reviewer 5 Report
EUV lithography, which plays a very important role in the semiconductor fabrication process below 10nm. Pellicles as a protective film of nm thickness needs to have extremely high characteristics in terms of thermal resistance, high transmittance, and less mechanical deflection. This manuscript reported a commercial tool based finite element method (FEM) and Python program based analytical-numerical method to discuss those properties of pellicle films.
Overall, this manuscript is timely and valuable work. Here are comments that need the authors to address.
1. Materials for commercial pellicles include poly-silicon(ASML), monocrystalline silicon(S&S tech), silicon carbide(FST), CNT(IMEC), graphite (Samsung). Can you explain the choice of materials used in the simulation (In table 2)
2. The authors have conducted a scientific comparison and analysis in the work of simulation part. Do other programming languages achieve the same purpose? Such as C /C++ /MATLAB.
3. The conclusion section at the end is too general and is recommended to be further streamlined.
4. Format of fig 3.4.5 7.8 needs further correction.
Author Response
EUV lithography, which plays a very important role in the semiconductor fabrication process below 10nm. Pellicles as a protective film of nm thickness needs to have extremely high characteristics in terms of thermal resistance, high transmittance, and less mechanical deflection. This manuscript reported a commercial tool based finite element method (FEM) and Python program based analytical-numerical method to discuss those properties of pellicle films.
Overall, this manuscript is timely and valuable work. Here are comments that need the authors to address.
1. Materials for commercial pellicles include poly-silicon(ASML), monocrystalline silicon(S&S tech), silicon carbide(FST), CNT(IMEC), graphite (Samsung). Can you explain the choice of materials used in the simulation(In table 2)
(Answer) Thank you.
One of the reasons for the choice of materials used in simulation is that experiment data and material data were published in research papers. CNT and graphite will be next topic.
2. The authors have conducted a scientific comparison and analysis in the work of simulation part. Do other programming languages achieve the same purpose? Such as C /C++ /MATLAB.
(Answer) Thank you.
I think that it is possible to transfer Python language to other programming languages.
3. The conclusion section at the end is too general and is recommended to be further streamlined.
(Answer) Thank you. The last sentence in conclusion was deleted and two sentences were added.
In 292th line in conclusion,
The maximum transverse deflection in an analytical-numerical method was magnified 1000 times constantly due to thickness reduction of 0.1 multiplication. The difference of transverse deflection due to pressure between results of FEM and an analytical-numerical method was increased at thickness t= 5.4 μm (t/L = 0.00054).
5. Format of fig 3.4.5 7.8 needs further correction.
(Answer) Thank you. Figures 3, 4, 5, 6, 7, and 8 were redrawn.

Figure 4

Figure 5

Figure 6

Figure 7

Figure 8

For the written english
(Answer) Thank you. This paper was revised two times by the company of written English. This is certificate of editing.

Round 2
Reviewer 3 Report
The revised manuscript addressed my concerns regarding the novelty. However, as I am really not in the simulation research area, I cannot tell whether this manuscript is publishable or not (even though I pick "accept" below).
Reviewer 5 Report
The author has provided a reasonable explanation for my 4 queries, and I have no further queries in this manuscript and recommend receiving this work.